# Interventional Radiological Management and Prevention of Complications after Pancreatic Surgery: Drainage, Embolization and Islet Auto-Transplantation

**DOI:** 10.3390/jcm11206005

**Published:** 2022-10-12

**Authors:** Cristina Mosconi, Maria Adriana Cocozza, Filippo Piacentino, Federico Fontana, Alberta Cappelli, Francesco Modestino, Andrea Coppola, Diego Palumbo, Paolo Marra, Paola Maffi, Lorenzo Piemonti, Antonio Secchi, Claudio Ricci, Riccardo Casadei, Gianpaolo Balzano, Massimo Falconi, Giulio Carcano, Antonio Basile, Anna Maria Ierardi, Gianpaolo Carrafiello, Francesco De Cobelli, Rita Golfieri, Massimo Venturini

**Affiliations:** 1Department of Radiology, IRCCS Azienda Ospedaliero-Universitaria di Bologna, 40138 Bologna, Italy; 2Diagnostic and Interventional Radiology Department, Circolo Hospital, ASST Sette Laghi, Insubria University, 21100 Varese, Italy; 3Department of Radiology, IRCCS San Raffaele Scientific Institute, San Raffaele School of Medicine, Vita-Salute University, 20132 Milan, Italy; 4Department of Diagnostic Radiology, Giovanni XXIII Hospital, Milano-Bicocca University, 24127 Bergamo, Italy; 5Diabetes Research Institute, IRCCS San Raffaele Scientific Institute, Vita-Salute University, 20132 Milan, Italy; 6Clinical Transplant Unit, Division of Immunology, Transplantation and Infectious Diseases, IRCCS San Raffaele Scientific Institute, Vita-Salute University, 20132 Milan, Italy; 7Division of Pancreatic Surgery, IRCCS, Azienda Ospedaliero Universitaria di Bologna, 40138 Bologna, Italy; 8Pancreatic Surgery Unit, Pancreas Translational & Clinical Research Center, IRCCS San Raffaele Scientific Institute, 20132 Milan, Italy; 9Department of General, Emergency and Transplants Surgery, Circolo Hospital, ASST Sette Laghi, Insubria University, 21100 Varese, Italy; 10Radiodiagnostic and Radiotherapy Unit, Department of Medical and Surgical Sciences and Advanced Technologies, University Hospital “Policlinico-Vittorio Emanuele”, 95123 Catania, Italy; 11Diagnostic and Interventional Radiology Department, Fondazione IRCCS Cà Granda Ospedale Maggiore Policlinico, 20122 Milan, Italy

**Keywords:** pancreatectomy, interventional radiology (IR), drainage, hemorrhage, embolization, islet of Langerhans transplantation

## Abstract

Pancreatic surgery still remains burdened by high levels of morbidity and mortality with a relevant incidence of complications, even in high volume centers. This review highlights the interventional radiological management of complications after pancreatic surgery. The current literature regarding the percutaneous drainage of fluid collections due to pancreatic fistulas, percutaneous transhepatic biliary drainage due to biliary leaks and transcatheter embolization (or stent–graft) due to arterial bleeding is analyzed. Moreover, also, percutaneous intra-portal islet auto-transplantation for the prevention of pancreatogenic diabetes in case of extended pancreatic resection is also examined. Moreover, a topic not usually treated in other similar reviewsas percutaneous intra-portal islet auto-transplantation for the prevention of pancreatogenic diabetes in case of extended pancreatic resection is also one of our areas of focus. In islet auto-transplantation, the patient is simultaneously donor and recipient. Differently from islet allo-transplantation, it does not require immunosuppression, has no risk of rejection and is usually efficient with a small number of transplanted islets.

## 1. Introduction

Pancreatic adenocarcinoma represents 85% of all pancreatic cancers and is the fourth most common cause of cancer-related deaths [1]. It has a very poor prognosis with a 5-year survival of 5%, which is mostly due to its aggressive biological nature and early local invasiveness. Surgical resection can be performed in about 15–20% of the patients at the time of diagnosis [2]. In the remaining patients with locally advanced or metastatic disease, palliative treatments are required [3,4,5,6,7,8,9]. Moreover, pancreatic surgery is one of the most complex surgical abdominal procedures, which is due to anatomical location of the pancreatic gland, close to vascular structures and the biliary and gastrointestinal system. Despite the improvement of the surgical technique, post-operative mortality ranges from 2 to 6% with a high incidence of surgical complications: approximately of 50% [10,11,12]. The main complications of pancreatic resection may be divided into two main categories: nonvascular and vascular, such as inflammatory collections, bile leakage (BL) and post-operative pancreatic fistula (POPF) in the first category and post-pancreatectomy hemorrhage (PPH), pseudoaneurysm and vascular thrombosis in the second one. Clavien and Dindo [13] proposed a classification for grading post-operative complication after pancreatic resection. Grade I is defined as any modification of the normal post-operative course without the need for pharmacological or surgical treatment. Grade II requires the pharmacological treatment. Grade III involves surgical, endoscopic or interventional radiological procedures. Life-threatening and intensive care unit management defines Grade IV, while patient death depicts Grade V. Relaparotomy is performed in 4 to 20% of the cases, with a mortality ranging between 13% and 60% [14,15]. The management of these complications is extremely complex, and for this reason, the cooperation of a multidisciplinary team is necessary, involving interventional radiologists, endoscopists and surgeons. In this scenario, the role of the interventional radiologist is crucial for the diagnosis and the treatment. Indeed, Interventional Radiology (IR) depicts a viable alternative to surgical treatment, resulting in a definitive treatment without the need for re-operation [16]. Several IR procedures, such as the percutaneous drainage of fluid collections due to pancreatic fistulas, percutaneous transhepatic biliary drainage (PTBD) in case of bile leakage and arterial transcatheter embolization (or covered stenting) in case of post-operative hemorrhage, constitute minimally invasive procedures with a consequent reduction in the hospital stay and a considerable decrease in morbidity and mortality [17,18]. In selected and experienced centers, IR is also involved in the prevention of pancreatogenic diabetes after extensive pancreatic resections through percutaneous intraportal islet auto-transplantation [19,20,21,22,23]. This pictorial review is focused on interventional radiological management and the prevention of complications after pancreatic surgery, providing an update of the current scientific literature.

## 2. Percutaneous Fluid Collection Drainage

Percutaneous drainage placement as the management of fluid collections and/or abscesses after pancreatic surgery [24,25] is standard practice worldwide. The most common cause is represented by post-operative pancreatic fistula, which is described by the International Study Group on Pancreatic Fistula as a measurable fluid collection with an amylase concentration greater than three times the serum amylase activity after three post-operative days [26]. Post-operative pancreatic fistula is the result of an injury of the pancreatic duct, which leads to a leak of pancreatic fluid and enzymes. Long-lasting fistulas often evolve in peri-pancreatic swelling, which may run into infection, hemorrhage and necrosis, leading to disconnected pancreatic duct syndrome. The incidence of fistulas after pancreatic resection varies between 9 and 29% [27,28], with a higher rate after a central pancreatectomy than that after a distal pancreatectomy [29]. Endoscopic retrograde cholangiopancreatography (ERCP) is the first-line therapy in the management of pancreatic fistulas, but it is not always feasible, especially in patients who have undergone gastroduodenal surgery [30].

In peri-pancreatic fluid collections, the use of percutaneous drainage using imaging as a guide will find a wide-ranging viability thanks to the higher rate of technical success, with a low rate of complications. Zink et al. reported a success rate of 97.6% regarding puncture techniques in percutaneous drainage after pancreaticoduodenectomy [31]. When the fluid collection is ultrasonically visible, not deeply located, ultrasound-guided percutaneous drainage placement in real time is generally preferred (Figure 1).

Ultrasound (US) is widely available, easy to handle, and free from radiations. US-guided percutaneous drainage performed in an angiographic room allows if necessary a second fluoroscopy phase to obtain a full collection opacification to demonstrate the pancreatic fistula or accidental communications with other structures (gastrointestinal, biliary, pancreatic). If the collection is deeply located in the abdomen and/or poorly visible at ultrasound, computed tomography (CT) represents a valid alternative to guide the safe placement of a percutaneous drainage (Figure 2).

US- or CT-guided percutaneous drainage procedure can be performed using two alternative techniques, the trocar or the Seldinger technique. The trocar technique is faster, useful in agitated patients, and based on a drainage catheter containing a trocar needle which is inserted directly into the collection. The Seldinger technique is safer, as indicated in the case of a small window and a long/difficult course: for example, in case of a retroperitoneal collection drained through an anterior approach to minimize possible complications [16]. The Seldinger technique is based on a puncture with a fine needle (for example, a 0.021 Chiba needle), advancement of a thin guidewire (for example, a 0.018 inch), progressive dilatation with different dilators, advancement of a 0.035/0.038-inch rigid guidewire, further dilatation with a 9/11 French dilation, and final placement of an 8–10 French pigtail (or curved) multi-hole catheter. The correct placement of the drainage can be confirmed by the aspiration of fluid material. In case of a complex collection composed by very dense/almost solid material, also larger catheters (14–16 French) may be used. Furthermore, the use of a dual-lumen drainage catheter with irrigation of a dedicated flush lumen may favor the evacuation of complex fluid collections [32,33]. Percutaneous fluid collection drainage is performed also in case of misplaced surgical drainages.

In the management of pancreatic fistulas, direct percutaneous pancreatic duct interventions may be considered as a temporizing therapy to prepare for elective surgery. Through direct duct cannulation, it is possible to drain the pancreatic juice at the origin and prevent the loss of enzymes in the peripancreatic space, reducing peri-pancreatic swelling, inflammation, and the extension of the fistula. Percutaneous pancreatic duct cannulation can be achieved through the puncture of peri-pancreatic fluid collection under combined US and fluoroscopic guidance. Subsequently, direct ductal interventions include the placement of pancreatic duct drainage, ruling out the injury and facilitating the recovery of the duct, the placement of stent or the embolization of the leak. In a way, the duct occlusion through embolization may simulate the effect of a partial pancreatectomy, solving the fistula [34].

The technical success rate of pancreatic duct cannulation, as shown in a recent study by Li et al. [27], was 90.9%. In the literature, the experience reported is narrow but showed that this kind of therapy should be considered in patients ineligible for surgery [35,36]. Moreover, endoscopic and percutaneous drainage showed a similarly high rate of clinical success, replacing surgery and reducing lengths of hospital stay [37,38,39,40,41,42,43].

## 3. Percutaneous Transhepatic Biliary Drainage

Percutaneous biliary procedures and particularly percutaneous transhepatic biliary drainage (PTBD) are widely used in the management of post-operative biliary leaks [44,45,46,47]. Bile leakage is defined, according to the International Study Group of Liver Surgery, as a fluid collection with an increased bilirubin concentration at least three times greater than the serum bilirubin concentration at the same time [48]. Bile leakage shows an incidence between 4 and 10%, after pancreatic surgery, and it occurs most often after billion-enteric anastomosis and rarely because of a direct injury of the bile duct [49,50]. Percutaneous transhepatic cholangiography (PTC) is indicated as an alternative to Endoscopic Retrograde Cholangiopancreatography (ERCP) that is not diagnostic or when it cannot be performed because of difficult anatomies. In patients treated by pancreaticoduodenectomy with Roux-en-Y bilio-enteric reconstruction, endoscopic access to the biliary tree is not feasible [16]. PTC consists of a diagnostic IR procedure performed through the percutaneous puncture of a peripheral biliary duct with a fine needle (21–22 Gauges) using a right intercostal (Figure 3) or left sub-xiphoid approach (Figure 4) [51].

The use of a combined US and fluoroscopy guidance, with the one-wall puncture of the peripheral biliary duct in real time using US, can reduce the number of puncture attempts, mean procedural time and peri-procedural complications incidence [52]. Anyway, the greatest technical challenge in PTC due to a biliary leak is the puncture of not dilated biliary ducts, which may be overcome with the support of US guidance. The peripheral portal branch which courses parallel to the biliary duct can be used as a target to try to puncture the biliary duct. Differently from the right non-dilated biliary ducts, left not-dilated biliary ducts may still be visible due to aerobilia: in both cases, an angle of puncture more parallel than perpendicular to the portal–biliary course can facilitate biliary duct cannulation. Once a biliary duct is accessed and opacified, a microwire can be advanced through the needle to secure percutaneous access. Biliary anatomy and a leakage site can be demonstrated with a cholangiogram. If an anastomotic leakage is demonstrated, after dilatation of the intrahepatic tract, an internal-external biliary drainage catheter can be placed passing enterohepatic anastomosis to drain the bile from the biliary tree to the bowel in order to also allow anastomosis healing. An external drainage catheter may also be placed in patients with suspected intestinal leakage [2]. The technical success of PTBD reported a high rate of success, as shown in a recent multicenter study with a technical success of 100% and clinical success of 78% [44]. Moreover, the therapeutic success rate of PTBD and of endoscopic biliary drainage is similar, even if it is important to specify that cholangitis and pancreatitis after PTBD are lower [53] and that an endoscopic approach is not always possible. PTBD can be left in place for 4 to 8 weeks to obtain leak resolution, performing multiple exchanges and controls to evaluate the catheter position and function. When the injury persists, other options may be considered, such as the placement of a stent graft or the employment of embolizing agents [54]. The use of retrievable covered stents is gaining a foothold in the last years, accordingly to numerous studies, with a high rate of clinical success, providing an effective and definitive treatment [55]. On the other hand, other options which may be taken in consideration are the employment of balloon occlusions [46] or embolizing materials, such as fibrin, coils, and acetic acid, with an increased use of the N-butyl cyanoacrylate, according to the literature [56,57,58].

## 4. Transcatheter Embolization (Stent–Graft)

Endovascular treatment by transcatheter embolization (TE) (or stent graft) represents the first-choice option in case of arterial bleeding following pancreatic surgery [59,60,61,62]. The incidence of post-pancreatectomy hemorrhage (PPH) has been reported in up to 10% of cases and between the complications shows the higher rate of mortality, ranging between 10 and 38% [63,64,65]. The International Study Group for Pancreatic Surgery developed a classification based on bleeding onset, location (intraluminal or extraluminal) and severity (grade A, B and C) [66]. Grade A presents an early onset and no major clinical impact; grade B is classified as early severe or delayed mild cases, which require transfusions, observation in the intermediate care unit, or intervention; grade C includes delayed and severe hemorrhage, requiring intervention. The onset timing in PPH is fundamental. Indeed, early and delayed PPH displays different etiologies, frequencies and treatment. Delayed post-pancreatectomy hemorrhage is the most common; it shows mainly arterial origin, with pseudoaneurysms formation in one-third of the cases [67,68,69]. Delayed PPH is often the result of vessel erosion in case of fistula, abscess, or inflammation [70,71]. Early PPH occurs within 24 h after surgery, while delayed PPH occurs many days later, with a median onset of 10–27 days [72]. The only risk factor identified for bleeding was the presence of a pancreatic leak in 50% of patients with delayed bleeding. Intraluminal PPH is defined as blood draining from the nasogastric tube, hematemesis, or melena, while extraluminal PPH is defined as blood draining in an intraabdominal location [73]. The initial signs of PPH may be sentinel bleeding, a self-limiting episode of bleeding with blood loss from the drainage, hematemesis, and melena, not conditioning hemodynamic instability. This particular manifestation, corresponding to grade B delayed and mild PPH, may precede the development of hemorrhagic shock [61]. In fact, sentinel bleeding precedes a severe PPH in 50–80% of patients and thus requires immediate diagnostic workup. When PPH occurs, decisions regarding the most diagnostic or therapeutic intervention are made on the patient’s hemodynamic status and CT findings. In hemodynamic unstable patients, urgent laparotomy can be indicated, while in hemodynamically stable patients, after CT scan evaluation demonstrating arterial bleeding and/or pseudoaneurysm formation, angiography is recommended, avoiding invasive abdominal surgery [74]. Different techniques are available to arrest the bleeding once the hemorrhage site has been identified. The most common vessels involved are firstly gastroduodenal artery (GDA), which is followed by common hepatic artery (CHA), left hepatic artery, dorsal pancreatic artery, gastric artery, and splenic artery [75]. TE is usually performed via transfemoral, using 4–5 French standard angiographic catheters (Cobra, Simmons, Bernstein, etc.) and coaxial micro-catheters over different guidewires. Once the target vessel is terminal, a proximal TE can be performed to stop the bleeding and if collaterals are detected, their embolization is recommended, in the so-called “sandwich” technique, in order to prevent re-bleeding [76]. Several permanent embolizing agents can be employed, such as polyvinyl alcohol particles, coils (Figure 5), EVOH-based non-adhesive liquid embolic agents, or N-butyl-cyanoacrylate; instead, transient embolization with resorbable materials, such as gelatine or fibrin sponge, allows the reperfusion of the treated region after a variable length of time [61]. Coils are probably the embolizing agents most used, usually alone but also combined for example with EVOH liquid agents to enhance their occlusive power [77].

The choice of embolic agents/devices also depends on their availability and operator confidence. The technical success of endovascular treatment of delayed PPH has been reported as between 58% and 100%, with a low rate of mortality compared to surgical treatment and supporting the use of IR (Table 1).

For the treatment of visceral pseudoaneurysms following pancreatic surgery, the use of stent–grafts is spreading, especially in the last years with the advent of increasingly advanced technology ensuring the blood flow and reducing complications such as liver abscess or liver failure [88,89,90]. In fact, stent–grafts can determine pseudoaneurysm exclusion preserving the target vessel patency [82,91,92]. Stent–grafts are usually placed via transfemoral, using six to eight French long curved introducers or guiding catheters at the origin of the celiac trunk (or superior mesenteric artery) to facilitate stent–graft advancement toward the pseudoaneurysms over rigid guidewires (Figure 6).

Different kinds of stent grafts are available. In our opinion, self-expandable, low-profile, without shape memory stent–grafts are more suitable for visceral pseudoaneurysms than balloon-expandable, rigid stent–grafts, particularly in case of the marked tortuosity of the target arteries [93]. Indeed, this technique depicts a suitable option with a lower rate of recurrent bleeding and complications compared to selective embolization [94,95]. Nevertheless, the tortuosity of visceral arteries makes the stent–graft option feasible only in selected cases [62]. The choice of the IR technique to control PPH is left to the discretion of the interventional radiologist, considering the vascular anatomy, bleeding site, available material, and operator experience [82,96]. A rare, reported complication after pancreatic surgery is the finding of stenosis/occlusion/thrombosis of the mesenteric/splenic/portal axis. Percutaneous endovascular treatment can be performed with balloon dilation or stenting in case of post-surgical vascular strictures [97], with thrombo-aspiration or thrombolysis in case of portal thrombosis [98]. The access to the portal system can be achieved through a transhepatic route using a right intercostal or left sub-xiphoid approach and a combined US/fluoroscopy-guided technique.

## 5. Intra-Portal Islet Auto-Transplantation

Percutaneous intra-portal pancreatic islet transplantation is a well-known, repeatable procedure which allows a β-cell replacement therapy through a liver islet engraftment, leading to insulin release and glycemic control restoration in patients affected by type 1 diabetes [99]. Allo-transplantation, in which isolated and purified islets from the cadaveric donor [100] are injected in the portal vein, can be performed in patients with long-term diabetes after kidney transplantation, maintaining an immunosuppressive regimen to simultaneously cure diabetes and chronic renal failure [101]. Moreover, it can be also performed alone, according to the Edmonton protocol based on the infusion of an adequate islet mass and on a steroid-free immunosuppressive regimen, in patients with brittle type 1 diabetes with preserved renal function [102]. Allo-transplantation represents a valid, less invasive alternative to surgical pancreas transplantation. Islet transplantation demonstrated significant beneficial effects in the prevention/stabilization of diabetic complications [103,104,105,106,107]. Auto-transplantation was more recently introduced not to cure type 1 diabetes but to prevent another type of diabetes known as “pancreatogenic diabetes” [19,20,21,22,23]. The origin of pancreatogenic diabetes is caused from an extreme disruption of glucose homeostasis after extensive pancreatic resection such as subtotal/total pancreatectomy for tumors or chronic pancreatitis [108,109,110,111,112]. The percentage of patients undergoing pancreatectomy that develops pancreatogenic diabetes varies from 8% to 23% increasing up to 40–50% during the follow-up [112,113]. Auto-transplantation is usually performed 12–48 h after pancreatic surgery, does not require immunosuppression and has obviously a lower rejection rate than allo-transplantation. Eligibility criteria for islet auto-transplantation in case of extensive pancreatic resection are established [19]. Patients submitted to auto-transplantation are both donors and recipients. After laparoscopic or open surgery, the pancreatic tissue is resected. In case of malignant tumor, one cm of the pancreatic remnant proximal to the pancreatic margin is resected and sent for frozen section examination to confirm margin negativity [19,20]. Islets are isolated, centrifugated, purified, and transplanted 12–48 h after the surgical treatment usually in an angiographic room as in allo-transplantation [114,115,116,117]. Percutaneous intra-portal islet auto-transplantation is preferentially performed using a combined US and fluoroscopy guidance to reduce the number of puncture attempts, mean procedural time and peri-procedural complications [118]. The puncture of a peripheral branch of the portal vein is usually performed with a fine needle (21–22 Gauge) in local anesthesia under US guidance through a right-sided intercostal approach. Fluoroscopic guidance is used to perform the main trunk portal vein catheterization, finally placing a 4 or 5 French catheter: isles are slowly injected to avoid their rupture or damage in 15–20 min [115,116,117,118]. Before and after islet transplantation, portography and portal pressure measurement are usually always performed (Figure 7).

At the end of the procedure, the catheter is slowly retracted, and intrahepatic tract embolization may be performed using different embolic agents (gelatin sponge pledgets, glue, coils) to prevent bleeding. Technically, allo- and auto-transplantation are similar procedures which are performed by interventional radiologists in the same way. Anyway, auto-transplantation is performed 12–48 h after pancreatic surgery: this peculiar situation may cause hemodynamic instability and a suboptimal ultrasound visibility due to residual abdominal air, which may hinder the procedure [21].

## 6. Conclusions

Interventional radiology plays a crucial role in the management and prevention of complications after pancreatic surgery. Complications such as fluid collections, biliary leaks and arterial bleedings can be effectively managed reducing recovery time and avoiding surgical re-treatment morbidity: percutaneous drainages, percutaneous transhepatic biliary drainages, transcatheter embolization (or stent–graft placements) for experienced interventional radiologists are feasible, safe and effective with a low complication rate. In selected centers, interventional radiologists can also prevent pancreatogenic diabetes through percutaneous intra-portal islet auto-transplantation in case of extensive pancreatectomy.

## Figures and Tables

**Figure 1 jcm-11-06005-f001:**
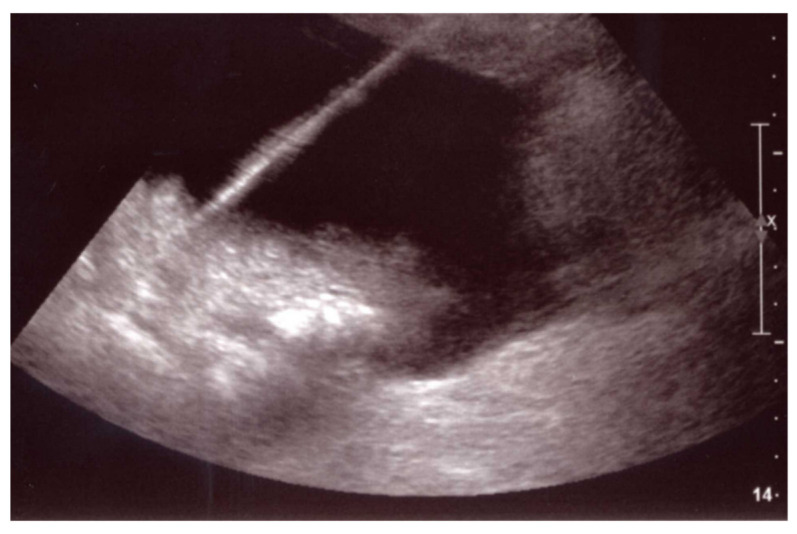
Fluid collection US-guided drainage. A percutaneous drainage can be easily placed in real time under US guidance in a fluid collection post-pancreatic surgery if it is superficially located and/or ultrasonically visible.

**Figure 2 jcm-11-06005-f002:**
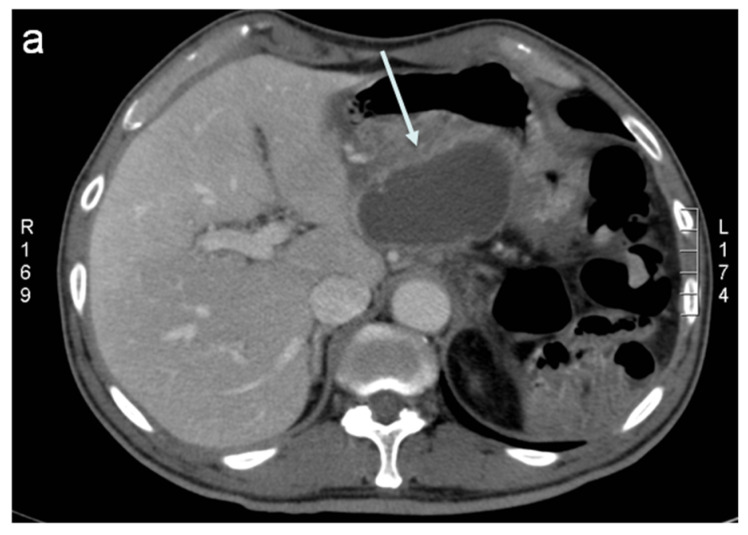
Fluid collection CT-guided drainage. (**a**) A contrast-enhanced CT showing a large, deep fluid collection after pancreatic surgery (arrow). (**b**) Patient positioning in the right lateral decubitus. (**c**) Percutaneous drainage (arrow) placed under TC guidance.

**Figure 3 jcm-11-06005-f003:**
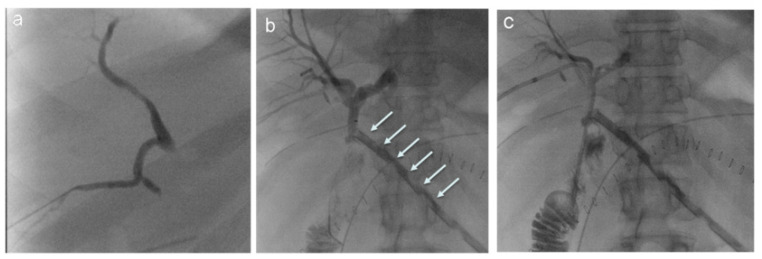
Biliary drainage using a right intercostal approach. (**a**) PTC performed through a puncture of a peripheral biliary duct of segment 6 in a patient with a biliary leak after pancreatic surgery. (**b**) PTC confirms the presence of a large biliary leak with a full opacification of the surgical drainage (arrows). (**c**) Using the same peripheral, right approach, an internal–external biliary drainage with extreme in the post-anastomotic jejunal loop is placed.

**Figure 4 jcm-11-06005-f004:**
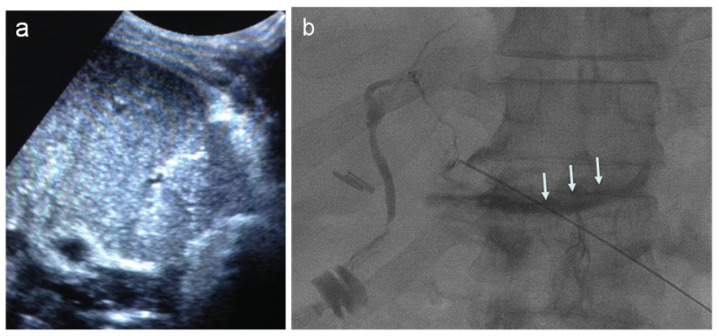
Biliary drainage using a left sub-xiphoid approach. (**a**) PTC performed through a puncture of a peripheral biliary duct of segment 3 in a patient with a biliary leak after pancreatic surgery. Biliary ducts are undilated and ultrasonically undetectable: anyway, the puncture is performed under US guidance parallel to the peripheral portal branch course to make the guidewire advancement in case of biliary tree opacification easier. (**b**) PTC demonstrating the extraluminal spreading of contrast material (arrows) due to a biliary leak.

**Figure 5 jcm-11-06005-f005:**
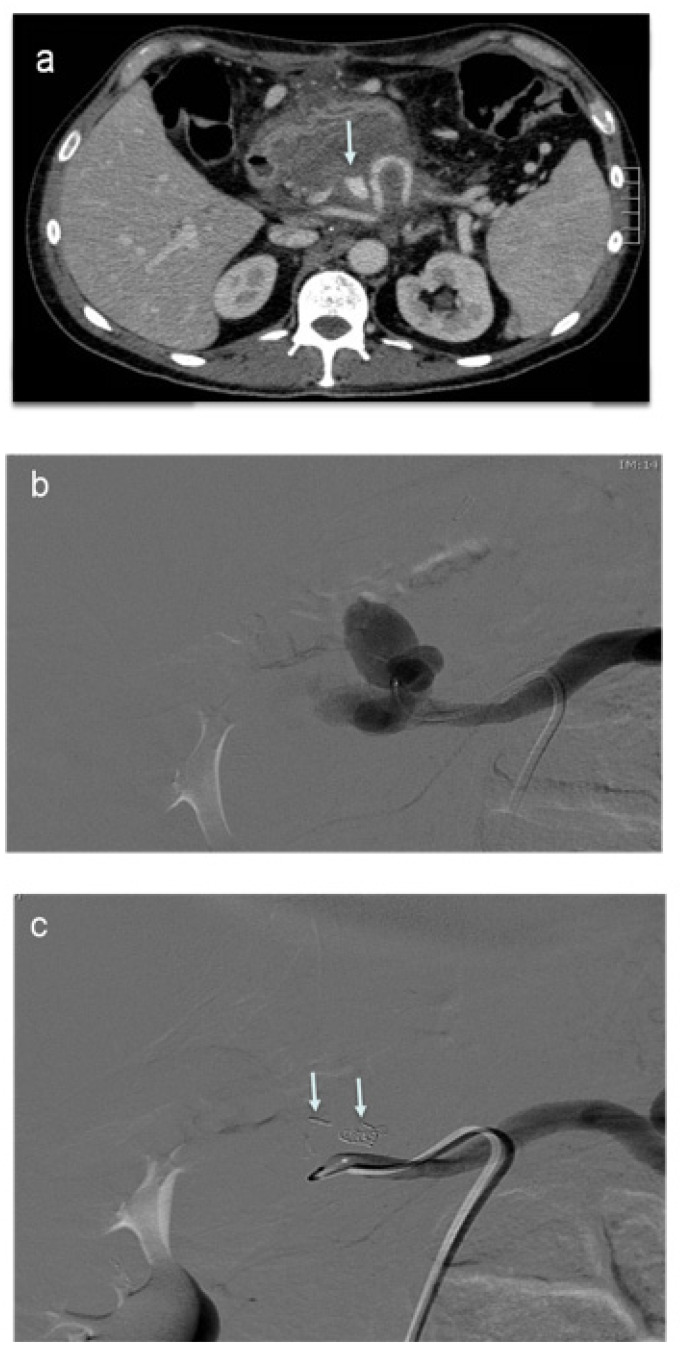
Splenic artery pseudoaneurysm embolization. (**a**) Contrast-enhanced CT after pancreatic surgery in a hemodynamically unstable patient shows a bleeding splenic artery pseudoaneurysm (arrow). (**b**) DSA showing a super-selective catheterization of the pseudoaneurysm using a coaxial microcatheter. (**c**) DSA showing a total pseudoaneurysm occlusion after embolization with coils (arrows).

**Figure 6 jcm-11-06005-f006:**
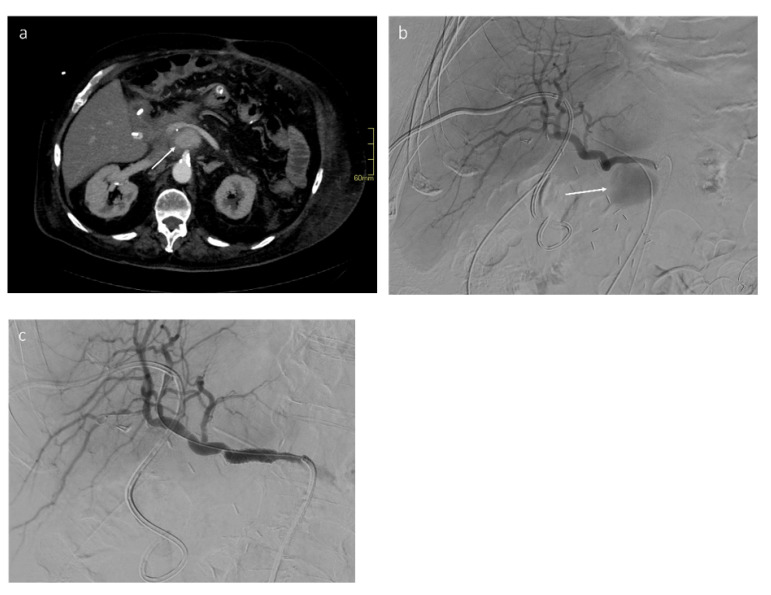
Bleeding pseudoaneurysm of the gastroduodenal artery stump. (**a**) Contrast-enhanced CT in arterial phase following pancreaticoduodenectomy shows a bleeding pseudoaneurysm of the gastroduodenal artery stump (arrow). (**b**) DSA confirms a bleeding pseudoaneurysm of the gastroduodenal artery stump (arrow). (**c**) After stent–graft placement, DSA shows a total pseudoaneurysm exclusion with regular patency of the stent and of the hepatic artery.

**Figure 7 jcm-11-06005-f007:**
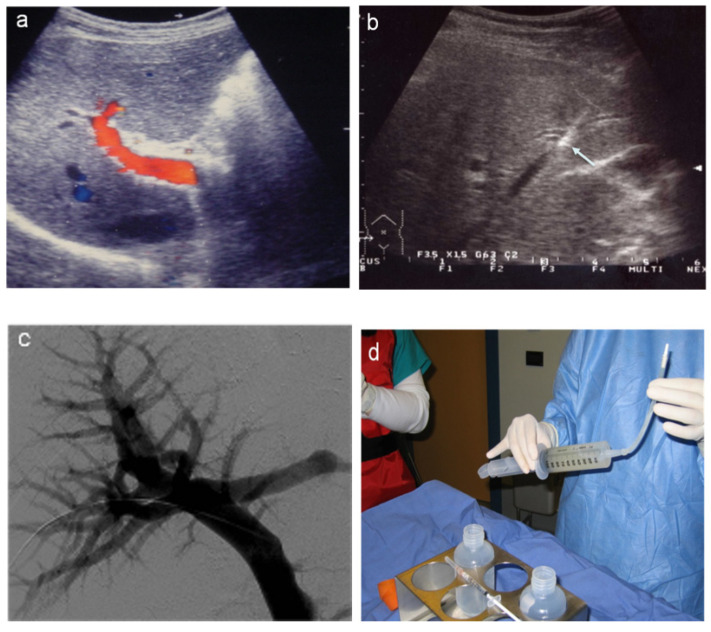
Percutaneous intra-portal islet auto-transplantation. (**a**) One day after extensive pancreatectomy, preliminary color Doppler examination showing portal vein patency. (**b**) Real-time, US-guided peripheral portal vein puncture (arrow). (**c**) After fluoroscopy-guided portal vein catheterization, preliminary portography, (**d**) before islet infusion.

**Table 1 jcm-11-06005-t001:** Outcome of endovascular treatments of bleeding after pancreatic surgery. DPH, delayed post-pancreatectomy hemorrhage; EPH, early post-pancreatectomy hemorrhage; E, embolization; S, stenting.

Study	Patients (n)	Timing of Hemorrhage	Type of Procedure	Technical Success (%)	Complications (%)	Recurrent Bleeding (%)	Mortality (%)
Zhang et al., 2020 [78]	15	DPH	E	89	15	23	30
Hasegawa et al., 2017 [79]	28	EPH + DPH	E + S	100	32	-	29
Ansari et al., 2017 [80]	10	EPH + DPH	E + S	80	-	-	-
Gaudon et al., 2016 [81]	42	DPH	E + S	69	12	28	13
Hassold et al., 2016 [82]	27	DPH	E + S	100	22	7	23
Ching et al., 2016 [83]	28	DPH	E + S	97	0	26	7
Asari et al., 2016 [84]	19	EPH + DPH	E	79	-	-	20
Ding et al., 2011 [85]	23	DPH	E + S	87	4	4	9
Bellemann et al., 2014 [86]	24	DPH	S	88	12	8	21
Yekebas et al., 2007 [87]	43	EPH + DPH	E	58	-	-	27

## Data Availability

No new data were created or analyzed in this study. Data sharing is not applicable to this article.

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
