# Peer review of "Interventional Radiological Management and Prevention of Complications after Pancreatic Surgery: Drainage, Embolization and Islet Auto-Transplantation"

_jcm, 2022, doi:10.3390/jcm11206005_

Round 1

Reviewer 1 Report

COMMENTS FOR THE AUTHOR:

Manuscript Number: JCM (ISSN 2077-0383)

The manuscript “Interventional radiological management and prevention of complications after pancreatic surgery: percutaneous fluid collection drainage, percutaneous transhepatic biliary drainage, transcatheter embolization and intra-portal islet auto-transplantation” by Cristina Mosconi and co-authors is a rather detailed and interesting review. The review has fairly covered essential aspects of interventional radiological management and complication post pancreatic surgery. Overall, it provides an update on the current scientific literature related to this area.

It is complete and well-written. I do not have main criticisms. However, the manuscript needs some refinement and correction of a few errors, as mentioned below:

 1). The title should be rephrased. The optimal choice of terms in the title, abstract, and keywords can increase the citation rates of your paper.

2). Abstract is very straightforward. Although it is clear regarding the highlighted points of review, it is still not describing the area of current research gap and future prospective on how this review is beneficial for readers and different from other related reviews published to date.

3)  Authors have preferred using long statements, which is not an issue if the meaning is clear. However, proper punctuation and conjunctions are often missing in long statements and rendered such statements very confusing and complicated. Try to make it simple and split the sentences wherever it is needed. Please check the proper spacing. It is desirable to proofread the manuscript.

Author Response

Reviewer 1

The manuscript “Interventional radiological management and prevention of complications after pancreatic surgery: percutaneous fluid collection drainage, percutaneous transhepatic biliary drainage, transcatheter embolization and intra-portal islet auto-transplantation” by Cristina Mosconi and co-authors is a rather detailed and interesting review. The review has fairly covered essential aspects of interventional radiological management and complication post pancreatic surgery. Overall, it provides an update on the current scientific literature related to this area.

It is complete and well-written. I do not have main criticisms. However, the manuscript needs some refinement and correction of a few errors, as mentioned below:

 1). The title should be rephrased. The optimal choice of terms in the title, abstract, and keywords can increase the citation rates of your paper.

  1. According to your suggestion, in the revised manuscript the title was rephrased and shortened as follows: ”Interventional radiological management and prevention of complications after pancreatic surgery: drainage, embolization and islet auto-transplantation”.

2). Abstract is very straightforward. Although it is clear regarding the highlighted points of review, it is still not describing the area of current research gap and future prospective on how this review is beneficial for readers and different from other related reviews published to date.

  1. According to your suggestion, in the revised manuscript the end of the abstract was expanded as follows: ” …Moreover a topic not usually treated in other similar reviews as percutaneous intra-portal islet auto-transplantation for the prevention of pancreatogenic diabetes in case of extended pancreatic resection is also focused. In islet auto-transplantation the patient is simultaneously donor and recipient. Differently from islet allo-transplantation, it doesn’t require immunosuppression, has no risk of rejection and is usually efficient with a small number of transplanted islets”.

3)  Authors have preferred using long statements, which is not an issue if the meaning is clear. However, proper punctuation and conjunctions are often missing in long statements and rendered such statements very confusing and complicated. Try to make it simple and split the sentences wherever it is needed. Please check the proper spacing. It is desirable to proofread the manuscript.

  1. According to your suggestion, we corrected punctuation and conjunctions throughout the text to make the meaning of long sentences clearer

Reviewer 2 Report

I congratulate the authors for the work done and to consider publishing “Review Interventional radiological management and prevention of complications after pancreatic surgery: percutaneous fluid collection drainage, percutaneous transhepatic biliary drainage, transcatheter embolization and intra-portal islet auto-transplantation”

Specific comments

Keywords should be acording to MESH terms (https://meshb.nlm.nih.gov/MeSHonDemand)

Page 6-

Line 202 Change “hepatico-entero anastomosis” to “enterohepatic anastomosis”

Line 221 correct “pf” to “of”

Page 7

What about the use of EVOH??

Page 8

Table 1: Please include in the table note What is the meaning of EPH, DPH, E, E+S, S etc. 

Page 9

First paragraph, please explain why autoexpandable are more suitable for visceral aneurysms, because autoexpandable covered stent usually are larger profile, needs larger sheath compared with balloon expandable.

Author Response

Reviewer 2

I congratulate the authors for the work done and to consider publishing “Review Interventional radiological management and prevention of complications after pancreatic surgery: percutaneous fluid collection drainage, percutaneous transhepatic biliary drainage, transcatheter embolization and intra-portal islet auto-transplantation”

Specific comments

1) Keywords should be according to MESH terms (https://meshb.nlm.nih.gov/MeSHonDemand)

  1. According to your suggestion, we corrected Keywords according to MESH terms : Pancreatectomy; interventional radiology (IR); drainage; haemorrhage; embolization; islet of Langerhans transplantation.

2)Page 6- Line 202 Change “hepatico-entero anastomosis” to “enterohepatic anastomosis”

  1. According to your suggestion, we Changed “hepatico-entero anastomosis” to “enterohepatic anastomosis”

3) Page 6- Line 221 correct “pf” to “of”

  1. According to your suggestion, we corrected “pf” to “of”

4)Page 7- What about the use of EVOH??

  1. According to your suggestion, in the revised manuscript EVOH was mentioned and the following sentence was added:” Once the target vessel is terminal, a proximal TE can be performed to stop the bleeding an if collaterals are detected, it is recommended their embolization, in the so-called “sandwich” technique, in order to prevent re-bleeding [76]. Several permanent embolizing agents can be employed, such as polyvinyl alcohol particles, coils (Figure 5), EVOH-based non-adhesive liquid embolic agents, or N-butil-cyanoacrylate; instead, transient embolization with resorbable materials, such as gelatine or fibrin sponge, allows the reperfusion of the treated region after a variable length of time [61]. Coils are probably the embolizing agents most used, usually alone but also combined for example with EVOH liquid agents to enhance their occlusive power (Venturini M, et al. Elective Embolization of Splenic Artery Aneurysms with an Ethylene Vinyl Alcohol Copolymer Agent (Squid) and Detachale Coils. J Vasc Interv Radiol 2020, 31:1110-1117).

5) Page 8- Table 1: Please include in the table note What is the meaning of EPH, DPH, E, E+S, S etc.

  1. According to your suggestion, in the caption we included the meaning of EPH, DPH, E, E+S, S etc.

6) Page 9- First paragraph, please explain why autoexpandable are more suitable for visceral aneurysms, because autoexpandable covered stent usually are larger profile, needs larger sheath compared with balloon expandable.

  1. I think that self-expandable stent-grafts are more suitable than balloon-expandable stent-grafts to the curvilinear course of the visceral arteries particularly due to the lack of the shape memory. In my opinion balloon-expandable stent-grafts are more rigid and suitable for short and straight arteries: in some cases, they seem to force the artery to adapt to the stent. However that sentence was changed as follows: “In our opinion self-expandable, low-profile, without shape memory stent-grafts are may be more suitable for visceral pseudoaneurysms than balloon-expandable, rigid stent-grafts, and more adaptable to navigate through the tortuous visceral arteries particularly in case of marked tortuosity of the target arteries [92]”.